# Distribution and Prevalence of Elbow Pain (EP) in Elite Swimmers in Tianjin, China—A Regional Epidemiological Study

**DOI:** 10.3390/healthcare11192612

**Published:** 2023-09-22

**Authors:** Weihan Li, Maryam Hadizadeh, Ashril Yusof, Mohamed Nashrudin Naharudin

**Affiliations:** Faculty of Sports and Exercise Science, Universiti Malaya, Kuala Lumpur 50603, Malaysia; s2027445@siswa.um.edu.my (W.L.); ashril@um.edu.my (A.Y.); nashrudin@um.edu.my (M.N.N.)

**Keywords:** elbow pain (EP), swimming, prevalence, distribution, overhead sports, Tianjin China

## Abstract

Elbow pain (EP) in overhead sports is a common phenomenon. Swimming is classified as an overhead sport, but a lack of attention regarding EP in swimming has created a gap in the knowledge around elite freestyle swimming in Tianjin, China. The purpose of this study was to identify the distribution and prevalence of EP among elite freestyle swimmers in Tianjin, China. The methodology involved a total of 311 qualified participants who volunteered to undertake all measurements. The main findings showed that 183 of the elite freestyle swimmers in this study had EP (accounting for 58.84% of the total 311 participants), with 147 in slight pain and 36 in critical pain. The characteristics of a heavier body weight, taller height, longer left/right forearm length, and longer weekly average training hours serve as contributing factors to the occurrence of EP issues. Gender, age, weekly average training hours, and left/right forearm length are the primary factors correlating with and influencing the assessment scores. In conclusion, swimmers with longer weekly training hours, older ages, heavier bodyweight, and longer forearm lengths should receive greater attention in relation to EP. Regular assessments at a high frequency serve as a means to identify the risk of EP.

## 1. Introduction

Elbow pain (EP) is a common sports injury in various overhead sports [1]. As it is considered an overhead sport, scholars have always been interested in swimming [2]. However, EP in swimming has been neglected due to the influence of people’s general cognition of sports injuries caused by swimming, with most believing that swimming injuries are concentrated in the shoulder, lower back, and knee and ankle joints [3]. Meanwhile, due to the influence of different continents and regions, attention to swimming events, EP in swimming has received a low level of attention for a long time. However, the United States, a sports powerhouse, has initiated relevant researches on EP in swimming in the last 10 years [4].

In China, research on and exploration of EP in swimming are in their initial stages [5]. As a popular sport, swimming attracts ordinary young people to become elite athletes [6]. In recent years, the total number of registered elite swimmers on the Chinese mainland has increased steadily, most recently totaling 87,886, but the number of athletes who have been forced to quit because of various overuse injuries has increased yearly [7]. Although shoulder injuries have the highest incidence rate, lower back pain and injuries of joints such as the knees, ankles, and elbows have been on the rise in recent years [5,8,9]. However, there is a serious lack research evidence on the issue of EP in elite freestyle swimmers. Therefore, assessing EP in this population is a necessary measure to reduce the risk of early retirement from sporting careers.

There are a few Chinese researchers who have initiated investigations into the condition of EP in swimming. Such studies are valuable for assessing the health status of elite swimmers’ upper limbs, highlighting specific risks that require attention, and providing information for targeted interventions aimed at preventing EP [1,5,10,11]. However, the majority of studies in the existing literature have primarily focused on exploring the relationship between the anthropometric variables of EP athletes (such as height, weight, gender, and forearm length) and their sports performance, as well as the relevance of these factors in participant selection [12,13,14,15,16]. The emphasis has been on investigating the impact of these variables and their connections with performance rather than delving into whether these factors directly contribute to the onset of EP [17,18]. This lack of investigation has hindered athletes and coaches from making informed decisions to facilitate the return of EP patients to the sport accordingly. A previous study investigated sports injuries in 194 elite swimmers in the city of Shanghai and the province of Zhejiang; its results showed that the incidence of EP was as high as 21% with an upward trend, ranking fourth among swimmers with a high incidence of sports overuse injuries [11]. This incidence of EP problems among freestyle swimmers is 7–13% higher than in the other three swimming-style athletes [11,19]. However, Tianjin, as one of the main training bases for elite swimmers in China, has been registering over 1000 swimming athletes annually since 2017 [20]. Unlike Shanghai and Zhejiang province, Tianjin has paid little attention to the impact of EP among this population [10,21]. Currently, China has not established an Injury Surveillance System (ISS) similar to the National Collegiate Athletic Association (NCAA) in the United States to monitor real-time data on various sports injuries [3]. This has resulted in a lack of coverage of elite swimming athletes throughout China in EP-related research, which must rely on regional sampling methods for investigation. In addition, the anthropometric characteristics of EP elite swimmers are essential data support for establishing a sports injury detection system in China. In summary, investigating EP problems among elite freestyle swimmers in Tianjin will fill the gap in the study of EP in swimming.

In summary, regional sampling evaluation is one of the primary methods for studying EP in swimming athletes, given the state of development of sports science in China. Regular assessment of athletes’ physiological parameters over time is a necessary injury prevention strategy to safeguard their health and professional safety [22,23]. Evaluating the anthropometric characteristics of EP elite swimmers will help establish a similar ISS system that can offer real-time monitoring of various sports injuries and provide data support in the near future. Thus, to the best of our knowledge, this is the first study to evaluate the distribution and prevalence of elbow pain (EP) among elite freestyle swimmers in Tianjin, China. The purpose of this study was to assess the number of swimmers with EP and their anthropometric characteristics in Tianjin, China, as well as the correlation between the anthropometric characteristics of EP swimmers and the assessment results, which will identify factors associated with EP.

## 2. Materials and Methods

This study received ethical approval from the University of Malaya Ethical Approvals Sub-Committee, with reference number UM.TNC2/UMREC_1951 (Appendix D). All subjects provided written consent before commencing this study (Appendix E). This study proceeded in accordance with the University of Malaya’s Research Ethics Guidelines.

### 2.1. Sample Recruitment

The subjects in this study were swimmers from three different competitive-level organizations (club, university, and professional swimming teams). This study was conducted from February to September 2022 in Tianjin Evergrande Olympic Aquatic Sports Center, China. A total of 503 elite freestyle swimmers responded to join this study voluntarily. Recruitment information was obtained by telephone and e-mail. Based on the inclusion and exclusion criteria, a total of 311 swimmers were enrolled after excluding 192 athletes.

The inclusion criteria were: (1) Achieved the title of elite swimmer in freestyle swimming (identified as a personal best (PB) time in 50 m freestyle swimming under 24.50 s for males and under 27.20 s for females) [10]; (2) aged from 18 to 28 years; (3) non-paralysis swimming athlete.

The exclusion criteria were: (1) At the stage between competition and/or retirement; (2) having suffered from any injuries in the last six months, such as fractures, impingements, surgery, or injuries during physical therapy.

### 2.2. Sample Size Consideration

No formal sample size calculations were conducted at the beginning of this study due to the pragmatic approach. The final sample size was decided based on a predefined time frame of seven months. This decision was made following the principle of convenience or opportunistic sampling [24], taking into account the number of swimming athletes available in Tianjin. Based on the results of a preliminary evaluation of the percentage of EP swimmers (approximately 58%), set with α = 0.05, μ**_α/2_** = 1.96, π = 0.58, and a permissible error of 10% (δ = 0.1π) in formula $N=\frac{\mu_{\alpha /2}^{2}\pi \left(1-\pi \right)}{\delta^{2}}$, as well as considering a dropout rate of 20%, the final sample size was approximately 309.

### 2.3. Data Collection

One of the researchers (WH.L.) evaluated all 311 participants in two steps, namely, basic demographic and anthropometry measurement (Appendix C) and an EP assessment. In the first step, participants were asked about their gender, age, height (cm), weight (kg), handedness, organization, average weekly training hours, left forearm length (mm), right forearm length (mm), and any discomfort in the elbow joint, which, if they answered “yes,” required the provision of specific evidence, including discomfort period or medical diagnosis certificate. In the second step, the Upper Limb Function Evaluation (ULFE) (Appendix B) and Body Part Disability Evaluation (BPDE) (Appendix A) were used to assess the number and distribution status of EP patients. The participants completed the assessments in sequence according to the given motions in the questionnaire and reported scores after each action, which were sorted out by the researcher.

The ULFE is a test that assesses whether participants have EP and the degree of pain based on elbow joint actions in 13 motion scenarios. The participants were instructed to perform the motions in the order specified by the ULFE. After completing each motion, they were then required to report their perceived pain scores to the researcher. Details of the self-marking in the scoring category can be found in Appendix B. Upon completion of the entire assessment, the total score for each subject was calculated to determine the classification of EP. In accordance with the ULFE scoring protocol, the participants self-scored their perceived elbow joint pain sensation after completing each given task. The scores ranged from 0 (no discomfort) to 4 (pain causing heart rate changes) (Appendix B). After completing all of the assigned tasks, the total score was calculated, and participants with a score exceeding 5 were classified as EP patients (Appendix B). Based on the pain category classification (Appendix B), the participants’ total scores were used to determine the pain categorization, with slight pain EP categorized as a total score between 0 and 30, moderate pain as a total score between 30 and 35, and critical pain as a total score over 35. The 13 motion scenarios in the ULFE were throwing motion (unilateral), pull down (unilateral), push-off action (bilateral), elbow initiation bending at 30 degrees, elbow initiation bending at 45 degrees, elbow passivity bending at 30 degrees, elbow passivity bending at 45 degrees, elbow initiation weighted bending at 30 degrees, elbow initiation weighted bending at 45 degrees, elbow passivity weighted bending at 30 degrees, elbow passivity weighted bending at 45 degrees, elbow abduction (unilateral), and loaded elbow abduction (unilateral).

The BPDE is a test that assesses whether participants have EP and the degree of pain based on elbow joint actions in 10 daily scenarios. The participants were instructed to perform the motions in the order specified by the BPDE. After completing each motion, the subjects were required to report their perceived pain scores to the researcher. The details of the self-marking in the scoring category can be found in Appendix A. Upon completion of the entire assessment, the total score for each subject was calculated to determine the classification of EP. In accordance with the BPDE scoring protocol, the participants self-scored their perceived elbow joint pain sensation after completing each given task. Scores ranged from 0 (no discomfort) to 4 (pain causing heart rate changes) (Appendix A). After completing all of the assigned tasks, the total score was calculated, and participants with a score exceeding 5 were classified as EP patients (Appendix A). Based on the pain category classification (Appendix A), the participants’ total scores were used to determine the pain categorization, with a total score between 0 and 25 categorized as slight pain, a total score between 25 and 30 as moderate pain, and a total score over 30 as critical pain. The 10 actions in daily scenarios were lift weights, take care of yourself, lift a weight and put it in place, move heavy objects, carry a plate or a tray, shower by self, touch your back or shoulder blades, get dressed, catch a flying saucer toy, and drop litter.

The ULFE and BPDE in this study are questionnaires designed based on the FMS scale to screen EP patients under both real-life and training conditions. According to the literature, the reliability of the Functional Movement Screen (FMS) scale is 0.81 (95% CI, 0.69–0.92) and 0.81 (95% CI, 0.72–0.92), respectively, while the odds ratios are within the 95% confidence limits for the injury predictive value [25,26,27,28,29,30]. However, no literature was found that assessed the reliability of the ULFE and BPDE. Therefore, this study examined the internal consistency of the two questionnaires. The results showed that Cronbach’s α coefficient for ULFE was 0.874 (α > 0.7) and for BPDE was 0.886 (α > 0.7), indicating good reliability for both questionnaires.

### 2.4. Statistical Analysis

SPSS 26.0 software was used for the data analysis. The Kolmogorov–Smirnov and Shapiro–Wilk tests were first used to determine all normality and homogeneity variance levels [31,32] (Appendix F). Box–Cox transformation was then used to normalize the data if they were not consistent with the normal distribution and homogeneity of variance [33]. Those variables that conformed to the normal distribution or had been normalized are represented using the mean and standard deviation (SD). The statistical tests were bilateral, and the difference was considered statistically significant when the *p*-value was less than 0.05.

The Pearson correlation method was used to assess the correlation between the UFLE and BPDE scores and all variables [34]. Then, multiple regression analysis was performed to assess the effect of the anthropometric and demographic variables on the UFLE and BPDE scores [35]. Finally, binary logistic regression was used to identify EP swimmers with 30 and 35 as the cut-off threshold in the UFLE and BPDE, respectively [36]. The *p*-value was set at less than 0.05.

## 3. Results

### 3.1. Descriptive Statistical Analysis Results of the 311 Subjects

All 311 eligible subjects completed all measurements. The demographic findings in Table 1 report that the number of EP patients exceeded half of the total number of participants (183 EP patients, accounting for 58.8%). The demographic results by gender indicate that the females were approximately one year younger than the males. Among all EP patients, the number of males was approximately three times that of females. According to gender and pain classification, the number of males with slight pain EP was approximately 3.5 times that of females. In the critical pain category, the proportion of males was approximately 1% higher than females. A total of three participants provided medical diagnosis certificates, which confirmed that they were suffering symptoms of “golfer’s elbow” (*n* = 1) and “tennis elbow” (*n* = 2), respectively.

Table 2 reports that the majority of the EP patients were from a professional swimming team (PST), accounting for 86.9%. Among them, the number of male EP patients was highest, which was approximately 2.5 times that of females. The average weekly training hours for females was approximately twice that of males. The duration of EP history was similar for both male and female EP patients, which exceeded two weeks. The difference in the 50 m freestyle swimming personal best (PB) time between male and female EP patients was approximately within 0.1 s.

### 3.2. Pain Category and the Anthropometric Characteristics of 183 EP Subjects

Based on the scoring categories of the UFLE and BPDE, the specific pain categories were determined for all 183 EP patients. Table 3 presents detailed information regarding these findings. The main results indicate that 147 EP patients were classified into the slight pain category and 36 EP patients into the critical pain category. No EP patients were categorized as having moderate pain.

Among the two pain categories and the EP patients’ anthropometric characteristics, it was found that female EP patients with critical pain had higher values than EP patients with slight pain in terms of age, height, and weight by approximately 0.5 years, 1 cm, and 0.5 kg, respectively. Male EP patients with critical pain had higher values than EP patients with slight pain in terms of age and height, but lower values in terms of weight, with the difference accounting for approximately 1 year, 0.5 cm, and 1 kg, respectively.

Both male and female EP patients with critical pain had longer left forearm lengths compared to those with slight pain, by approximately 2 and 8 mm, respectively. However, when comparing the right forearm length, female EP patients with critical pain had longer lengths than those with slight pain, by approximately 6 mm, while male EP patients with critical pain had shorter lengths than those with slight pain, by approximately 1 mm. All EP patients with critical pain were right-handed (accounting for 100%), while only four in five of EP patients with slight pain were right-handed.

### 3.3. Correlation between the UFLE and BPDE Scores and All Variables in 183 EP Subjects

The results of the Pearson correlation analysis (Table 4) indicate that among all variables, gender, weight, organization, and left and right forearm lengths had statistical significance, with *p*-values of less than 0.05. The correlation between gender and UFLE and BPDE scores yielded *r*-values of −0.299 and −0.152, indicating negative correlations. The correlation between weight and BPDE scores was positive, with an *r*-value of 0.146. Organization and left and right forearm lengths demonstrated positive correlations with the ULFE and BPDE scores, with *r*-values of 0.648/0.725, 0.409/0.515, and 0.399/0.514, respectively.

#### Effect and Predictors of the Anthropometric and Demographic Variables on the UFLE and BPDE Scores

The results of the multiple regression analysis (Table 5) indicate that, among all anthropometric and demographic variables, gender, age, right forearm length, average weekly training hours, and organization had statistical significance, with *p*-values of less than 0.05. Gender exhibited a negative correlation with the ULFE and BPDE scores, with β-values of −21.649 and −13.823, respectively. The effects of age, average weekly training hours, organization, and right forearm length showed positive correlations with the ULFE and BPDE scores, with β-values of 0.731/0.484, 0.607/0.572, 9.605/11.431, and 0.518/0.521, respectively.

The elbow pain predictor results of the binary logistic regression (Table 6) indicate that among all variables, gender and left/right forearm length had statistical significance, with *p*-values of less than 0.05. The odds ratio (OR) for the gender factor in ULFE scores ≥ 35 (critical pain category) and BPDE scores ≥ 30 (critical pain category) as the cut-off threshold was calculated to be less than 1 (OR = 0.376), which indicates a protective factor. This suggests that male swimmers tend to have lower assessment scores and lean toward the “slight pain” category. For the left/right forearm length factor, calculated in ULFE scores ≥ 35 (critical pain category) and BPDE scores ≥ 30 (critical pain category) as the cut-off threshold, the ORs were both greater than 1 the (OR = 1.168/1.177), indicating a risk factor. This implies that the longer the forearm length of EP swimmers, the more severe the intensity of their pain sensations.

## 4. Discussion

In this study, we obtained precise results regarding the number of EP patients, pain categorization, anthropometric characteristics, and distribution among elite freestyle swimmers in Tianjin, China. We also examined the contributing factors to EP. The observed high prevalence rate of EP (accounting for 58.8% of the total subjects) raises concerns about the occupational health development of elite swimmers in China.

The significant majority of male compared to female EP patients among the total number of 183 EP patients indicates that a significant number of male elite swimmers in China may be struggling with EP without finding appropriate solutions. This phenomenon also highlights the current state of occupational health development among Chinese swimmers, which exhibits a gender disparity with better occupational health levels observed among female athletes [37]. Therefore, our recommendation is that coaches need to pay attention to the occupational health development of male swimmers. Providing more opportunities for communication during daily training would allow these individuals to express their requirements and concerns more effectively [38]. This means that once an athlete experiences any discomfort, the coach can address it promptly and make a correct decision to prevent their EP from worsening.

This study’s evidence of a correlation between age, height, weight, right forearm length, and organization with the scores of the two EP assessments (UFLE and BPDE) indicates that the root cause of EP problems may lie within these factors. Regarding age, healthy athletes tend to be younger than EP patients. This suggests that the occupational health level of athletes declines with increasing age [39]. Therefore, we recommend that coaches should provide more sports protection measures and rest time for older athletes. Compared to healthy athletes, EP patients exhibit the anthropometric characteristics of shorter height, lighter weight, and shorter right forearm length. This indicates that athletes who possess these three features are more likely to experience EP problems than others. Therefore, during the athlete selection phase, such as teenagers between the ages of 15 and 18, relevant professionals should exclude individuals with these three anthropometric features or advise them not to participate in competitive swimming training to reduce the risk of EP [40]. In terms of organizations, although a certain number of EP patients come from university teams (UTs), professional swimming teams (PSTs) remain the primary hotbed for the greatest number of EP patients. Their training management strategies may have deficiencies, leading to a lack of timely and effective protection and intervention measures for EP swimmers to return to the sport promptly. Therefore, we suggest that swimming teams and clubs at all levels should consider implementing “periodization theory” as the training management principle to address deficiencies, which can provide athletes with maximum rest during busy training schedules [41]. Finally, ligament and tendon injuries could also be among the causes of EP [42,43,44,45]. In some clinical medical studies and anatomical literature, baseball players with elbow injures typically require surgical rehabilitation [44,46]. Therefore, for EP swimmers identified as experiencing critical pain in this study, we recommend further medical examinations to mitigate the risk of ligament and tendon injuries exacerbating their EP condition.

The results of pain categorization in EP patients show that the number of subjects with slight pain was approximately four times that of those with critical pain. This suggests that these subjects did not promptly report their EP and did not receive effective intervention measures to alleviate their EP discomforts. Therefore, we recommend that coaches and relevant professionals conduct frequent assessments of the occupational health of swimming athletes, aiming to detect and treat EP issues at an early stage to prevent the risk of deterioration [47]. Furthermore, with regards to the anthropometric characteristics of EP patients in relation to pain categorization, it was observed that the right forearm length of EP patients with slight pain was shorter than that of those with critical pain. This suggests that forearm length may impact the pain classification of EP patients, with a shorter forearm length being associated with lower levels of EP pain [48,49]. Therefore, based on the EP assessment scoring criteria, we recommend that EP patients with slight pain should undergo unloaded or lightly loaded upper limb strength training, such as isometric training, to improve arm muscle strength and endurance around the elbow joint, as an intervention method rather than relying solely on medication [50,51,52]. For EP patients with critical pain, they should exercise caution when selecting potential intervention methods. Upper limb strength training remains essential, but if the patient experiences any discomfort during the process, it should be immediately discontinued, and EP assessment or medical examination, such as magnetic resonance imaging (MRI), should be performed to avoid the risk of further injury [53].

In summary, despite the prevalence of EP issues among elite freestyle swimmers in Tianjin, China, potential solutions to reduce the number and risk of EP cases include improving training management strategies, optimizing athlete selection criteria, increasing the frequency of athlete occupational health assessments, and providing timely rest and intervention measures. Additionally, the anthropometric data of EP patients can support the establishment of a sports injury detection system in Chinese swimming teams in the near future.

## 5. Limitations

The sample inclusion and exclusion criteria may have overlooked some subjects who also suffer from EP, such as butterfly, backstroke, breaststroke, and synchronized swimmers. Additionally, young athletes under the age of 18 and disabled athletes were not included in the evaluation scope. Consequently, results based on such inclusion and exclusion criteria are preferred. Furthermore, factors such as ligament and tendon injuries, as well as shoulder strength, may contribute to EP among swimmers. These aspects were not investigated in the current study, potentially accounting for variations in the results. Therefore, it is advisable for future research to consider the correlation between these factors and elbow pain.

## 6. Conclusions

The high prevalence of EP issues among elite freestyle swimmers in Tianjin, China, necessitates increased attention from professionals. Special focus should be directed toward young male athletes within the EP swimmer cohort. Regular proactive assessments would benefit these swimmers. Organizational level and training duration emerged as pivotal factors impacting the elbow joint health of swimmers. Athletes of older age, longer forearm length, higher body weight, and taller stature should be prioritized for observation and assessment, and these traits characterize the EP swimmer profile. Finally, it is important to acknowledge the limitations of this study, including the limited sample range and the technical diversity of swimming. We acknowledge that the current inclusion criteria excluded young athletes (under 18 years old) and disabled athletes, and it is unknown whether EP exists in swimmers primarily practicing backstroke, breaststroke, and butterfly. Therefore, we plan to address these topics in future research.

## Figures and Tables

**Table 1 healthcare-11-02612-t001:** Demographics.

Demographics	Female	Male	Total
Gender distribution ^a^	98 (31.5%)	213 (68.5%)	311
Age (years) ^b^	18.85 (1.6)	20.84 (2.00)	-
EP patients’ gender distribution ^c^	49 (15.8%)	134 (43.1%)	183
Slight pain category’s gender distribution ^d^	33 (10.6%)	114 (36.7%)	147
Critical pain category’s gender distribution ^e^	16 (5.1%)	20 (6.4%)	36

Notes: Data are rounded to one decimal place. ^a^ All values are presented as *N* (% of total population). ^b^ All values are presented as mean and standard deviation (SD). ^c,d,e^ All values are presented as *N* (% of total population).

**Table 2 healthcare-11-02612-t002:** Distribution characteristics of organization, personal best (PB) completion time in 50 m freestyle swimming, average weekly training hours, and EP history by gender within all EP patients (*n* = 183).

Prevalence Characteristics		Female	Male
Organization ^a^	Club member (CM)	0	0
	University team (UT)	2 (1.9%)	22 (12.0%)
	Professional swimming team (PST)	48 (26.2%)	111 (60.7%)
PB (seconds (s)) ^b^		26.44 (0.6)	23.67 (0.5)
Average weekly training hours (hours) ^c^		25.41 (6.0)	11.95 (6.3)
EP history (weeks) ^d^		2.99 (2.4)	2.55 (2.4)

Notes: Data are rounded to one decimal place. ^a^ All values are presented as *N* (% of all EP patients). ^b,c,d^ All values are presented as mean and standard deviation (SD).

**Table 3 healthcare-11-02612-t003:** Anthropometric characteristics of the EP patients by gender (details of the EP pain categories).

Anthropometric Characteristics	Gender	EP Patients with Slight Pain (*n* = 147)	EP Patients with Critical Pain (*n* = 36)
Age (years) ^a^	Female	18.7 (0.5)	19.0 (0)
	Male	19.6 (1.4)	20.6 (0.5)
Height (cm) ^b^	Female	173.7 (3.4)	174.9 (3.9)
	Male	187.2 (3.4)	187.9 (1.5)
Weight (kg) ^c^	Female	57.7 (2.9)	58.0 (2.7)
	Male	77.60 (5.0)	76.5 (3.0)
Left forearm length (mm) ^d^	Female	249.8 (10.1)	257.6 (8.6)
	Male	279.9 (7.0)	281.7 (4.4)
Right forearm length (mm) ^e^	Female	251.5 (10.2)	257.5 (14.0)
	Male	283.2 (6.9)	282.1 (4.6)
Right handedness (individual) ^f^	Female	28 (19.1%)	16 (44.4%)
	Male	99 (67.4%)	20 (55.6%)
Left handedness (individual) ^g^	Female	5 (3.4%)	0
	Male	15 (10.2%)	0

Notes: Data are rounded to one decimal place. ^a–e^ All values are presented as mean and standard deviation (SD). ^f,g^ All values are presented as *N* (% of all EP patients).

**Table 4 healthcare-11-02612-t004:** Correlation between the ULFE and BPDE scores and all variables in the 183 EP subjects.

Variables	*r* in ULFE Scores	*p*-Value	*r* in BPDE Scores	*p*-Value
Gender	−0.299	<0.001	−0.152	0.041
Age	0.048	0.516	0.052	0.485
Height (cm)	−0.135	0.069	−0.011	0.878
Weight (kg)	0.001	0.986	0.146	0.048
Handedness	−0.03	0.683	−0.019	0.800
Average weekly training hours	0.055	0.457	0.081	0.274
Organization	0.648	<0.001	0.725	<0.001
Left forearm length (mm)	0.409	<0.001	0.515	<0.001
Right forearm length (mm)	0.399	<0.001	0.514	<0.001

Notes: *r-* and *p*-values are rounded to three decimal places.

**Table 5 healthcare-11-02612-t005:** Effect of anthropometric and demographic variables on the UFLE and BPDE scores.

Variables	ULFE Scores		BPDE Scores	
	β	*p*-Value	β	*p*-Value
Gender	−21.649 (−26.05 to 17.248)	<0.001	−13.823 (−17.67 to −9.976)	<0.001
Age	0.731 (0.48–0.982)	<0.001	0.484 (0.265–0.704)	<0.001
Height(cm)	−0.06 (−0.31 to 0.19)	0.638	−0.061 (−0.279 to 0.158)	0.583
Weight (kg)	0.136 (−0.125 to 0.398)	0.305	0.036 (−0.193 to 0.264)	0.758
Handedness	−1.744 (−4.207 to 0.719)	0.164	−0.958 (−3.111 to 1.195)	0.381
Average weekly training hours	0.607 (0.388–0.826)	<0.001	0.572 (0.38–0.763)	<0.001
Organization	9.605 (6.23–12.98)	<0.001	11.431 (8.481–14.382)	<0.001
Left forearm length (mm)	−0.016 (−0.263 to 0.232)	0.900	−0.122 (−0.339 to 0.094)	0.265
Right forearm length (mm)	0.518 (0.298–0.737)	<0.001	0.521 (0.329–0.713)	<0.001

Notes: β- and *p*-values are rounded to three decimal places.

**Table 6 healthcare-11-02612-t006:** Predictors of elbow pain.

Variables	ULFE Scores ≥ 35 and BPDE Scores ≥ 30	
	OR (95% CI)	*p*-Value
Gender	0.376 (0.176–0.805)	0.012
Age	0.911 (0.789–1.051)	0.202
Height (cm)	0.98 (0.924–1.039)	0.494
Weight (kg)	1.035 (0.995–1.077)	0.089
Handedness	0.653 (0.219–1.948)	0.444
Average weekly training hours	1.041 (0.913–1.186)	0.550
Organization	-	-
Left forearm length (mm)	1.168 (1.108–1.23)	<0.001
Right forearm length (mm)	1.177 (1.114–1.243)	<0.001

Notes: OR and *p*-values are rounded to three decimal places.

## Data Availability

Raw and processed data are available upon request to the corresponding author.

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
