# Peer review of "Distribution and Prevalence of Elbow Pain (EP) in Elite Swimmers in Tianjin, China—A Regional Epidemiological Study"

_healthcare, 2023, doi:10.3390/healthcare11192612_

Round 1
Reviewer 1 Report
The abstract, according to the journal guidelines, should be 200 words. Also, after the corrections revise it.
44. This is not the appropriate reference to describe the sentence. Please, recheck it.
Also, references 3 and 6 aren’t in any database. You use references to connect swimming with EP, but I did not notice everything about it.
52. Why only the freestylers? Both butterfliers and breaststrokers have serious issues with elbows in unpublished data.
81. It is a highly specialized research field to focuses solely on the freestyle style. I suggest expanding the scope to include other styles, such as backstroke, breaststroke, and butterfly. Have you considered the factors that may lead to EP (Elbow Pain)? Its cause could be attributed to overuse during training or the presence of a weak shoulder, which can result in unstable strokes and consequently place biomechanical stress on the elbow.
108-110. After the numbering, you start the sentence both with upper and lower case. Please, choose one way. Also, it could be more valid to use World Aquatic points to define the level of swimmers.
116. Please, explain how you used the term pragmatical strategy. In other instances, by conducting a G* power analysis, you can ensure the number of participants.
181. “p - value”
208. Use standard deviation (SD)
236. Why do you use mm instead of cm?
242. p
248. Use one color.
271,273. Choose one color
292. As I pointed out before, I think that the level of strength, and more specifically, the shoulder strength perhaps, contributes to this situation. Is there any literature that can regard this?
348. Write the limitations of the study in a separate section. Also, include the identification of strength level.
367. I suggest writing the tables in the Appendix, only in English. If the questionnaires are not validated in English, note under the table that are a translation for the needs of publication.
Check the whole literature. Especially, the introduction needs more appropriate references.
Overall, the manuscript does not contain many English issues, but it is important to carefully revise the introduction.
Author Response
please check out the attached file, which is the point-by-point response to reviewer 1 comments.

Reviewer 2 Report
The authors measured subjective ratings and demographics of swimmers with elbow pain and also measured anthropometric parameters to show their relationship. The goal and methods set in this study are clear, however, there are many weak points that prevent the current manuscript from being published. Specifically, the tables must be improved. The following are major concerns:
- Although ULFE and BPDE are major measures used in this study, how they were developed and evaluated was not explained in the manuscript. Furthermore, the relevant reference was not involved (l. 132-133). For example, the reviewer searched ULFE (upper limb function evaluate) and could not find the relevant sources. If the reliability of these measures is not supported, most of the current findings can not also be scientifically supported.
- the ULFE and BPDE attached in the manuscript would be better to be provided only in English for international readers; by translating the Chinese into English.
- The abbreviation of ULFE and BPDE was mentioned twice (l. 127, l. 190).
- Contents of the tables are hardly recognized, specifically, Table 2. The authors must rearrange the size and design of the tables.
- The authors used the term EP (elbow pain), however, the cause of joint pain varies; for example, one might be due to ligament damage, or on might be due to tendon injury, etc. The authors should more focus on the detailed discussion regarding the cause of the elbow pain.
- While Table 4 shows the effects of right forearm length on the ULFE and BPDE scores, Table 5 shows both forearm lengths are predictors of elbow pain. The authors should explain and discuss why only right forearm length was related to general ULFE and BPDE scores.
- In discussion, the phrase 'dominated by Yin and declining by Yang' cannot be understood and seems unnecessary to help comprehension of the readers in the world.
The quality of English used in this manuscript is not acceptable.
Author Response
please check out the attached file, which is the point-by-point response to reviewer 2 comments.

Round 2
Reviewer 1 Report
The manuscript was improved significantly. However, I still detected some flaws in the literature. The authors deleted ref. 3 and 6, however, I did not observe any additional references.
It is improved
Author Response
please checkout attached documents of point-by-point response to the reviewer's comments.

Reviewer 2 Report
The authors have reflected the comments given in the previous round quite well. For the current version of the manuscript, there are only a few minor comments.
In section 2.3 data collection
1. Please specify what is FMS scale.
2. The paragraph of l.126-134 might be better placed below the paragraphs of ULFE and BPDE introduction.
3. In Appendix 3, Kolmogorov–Smirnova to Kolmogorov–Smirnov
The current version was easy to review.
Author Response
please checkout attached document of point-by-point response to the reviewer's comments.
